# In vivo transcriptomic profiling using cell encapsulation identifies effector pathways of systemic aging

Omid Mashinchian[1,2†], Xiaotong Hong[1,3†], Joris Michaud[1‡], Eugenia Migliavacca[1‡], Gregory Lefebvre[1], Christophe Boss[1], Filippo De Franceschi[1], Emmeran Le Moal[4], Jasmin Collerette-Tremblay[4], Joan Isern[3], Sylviane Metairon[1], Frederic Raymond[1], Patrick Descombes[1], Nicolas Bouche[1], Pura Muñoz-Cánoves[3,5], Jerome N Feige[1,2*§], C Florian Bentzinger[1,4*§]

[1]Nestlé Institute of Health Sciences, Nestlé Research, Lausanne, Switzerland; [2]School of Life Sciences École Polytechnique Fédérale de Lausanne (EPFL), Lausanne, Switzerland; [3]Vascular Pathophysiology Area, Centro Nacional de Investigaciones Cardiovasculares, Madrid, Spain; [4]Département de Pharmacologie-Physiologie, Institut de Pharmacologie de Sherbrooke, Centre de Recherche du Centre Hospitalier Universitaire de Sherbrooke, Faculté de Médecine et des Sciences de la Santé, Université de Sherbrooke, Sherbrooke, Canada; [5]Department of Experimental and Health Sciences, Pompeu Fabra University, CIBERNED and ICREA, Barcelona, Spain, Barcelona, Spain

**\*For correspondence:**
jerome.feige@rd.nestle.com
(JNF);
cf.bentzinger@usherbrooke.ca
(CFB)

[†]These authors contributed
equally to this work
[‡]These authors also contributed
equally to this work

[§]These authors jointly directed
this work

**Competing interest:** See page
14

**Reviewing Editor:** Veronica
Galvan, UT Health San Antonio,
United States

**Abstract** Sustained exposure to a young systemic environment rejuvenates aged organisms and promotes cellular function. However, due to the intrinsic complexity of tissues it remains challenging to pinpoint niche-independent effects of circulating factors on specific cell populations. Here, we describe a method for the encapsulation of human and mouse skeletal muscle progenitors in diffusible polyethersulfone hollow fiber capsules that can be used to profile systemic aging in vivo independent of heterogeneous short-range tissue interactions. We observed that circulating long-range signaling factors in the old systemic environment lead to an activation of Myc and E2F transcription factors, induce senescence, and suppress myogenic differentiation. Importantly, in vitro profiling using young and old serum in 2D culture does not capture all pathways deregulated in encapsulated cells in aged mice. Thus, in vivo transcriptomic profiling using cell encapsulation allows for the characterization of effector pathways of systemic aging with unparalleled accuracy.

## Editor's evaluation

The manuscript includes in vivo studies where encapsulated myogenic progenitors are exposed to the systemic environment of young or aged mice. The authors provide a very important comparison of a novel approach to the use of young or aged serum in vitro, which is considered the current gold standard. The studies reported also provide evidence that the in vivo capsule-based method may constitute an alternative and possibly improved approach to the study of impact of environment-related changes on function of skeletal muscle progenitors.

## Introduction

Systemic cross-talk between tissues has emerged as an important determinant of organismal aging (*Demontis et al., 2013*). Supporting this notion, several features of tissue aging can be slowed or

reversed by heterochronic parabiosis (*Bert, 1864*; *Conboy et al., 2015*). Restoration of a youthful systemic environment has been shown to improve the function of muscle, heart, liver, brain, and other tissues in aged mice (*Baht et al., 2015*; *Conboy et al., 2005*; *Katsimpardi et al., 2014*; *Loffredo et al., 2013*; *Ruckh et al., 2012*; *Sinha et al., 2014*; *Smith et al., 2015*; *Villeda et al., 2011*). Interestingly, the positive effects of young blood on aged tissues appear to be milder than the pronounced negative effects of aged blood on young tissues (*Rebo et al., 2016*). This observation suggests the presence of dominant pro-aging factors in the systemic circulation, which cross-talk with local tissue niches to induce the global decline in organ function associated with old age. Recent studies have aimed at identifying the circulating factors involved in systemic aging, and some of the experimental interpretations have led to considerable controversy in the field (*Conese et al., 2017*).

It has long been known that a range of endocrine hormones are perturbed in later stages of life. Aging affects the somatotroph axis leading to decreased levels of growth hormone and insulin-like growth factor 1 (IGF-1) (*Garcia et al., 2000*). Levels of sex hormones such as testosterone and estrogen drop in aged individuals (*Medicine, 2004*). Old age is also associated with increased systemic inflammation that often goes along with a metabolic syndrome attributed to insulin resistance and an excessive flux of fatty acids (*Dominguez and Barbagallo, 2016*; *Franceschi and Campisi, 2014*). Next to these broad biological processes, several distinct pro- and anti-aging factors have been identified in the systemic environment. These include growth differentiation factor 15 (GDF15), eotaxin, β2-microglobulin, and transforming growth factor-β, which negatively affect brain or muscle tissue in aging (*Lehallier et al., 2019*; *Smith et al., 2015*; *Tanaka et al., 2018*; *Villeda et al., 2011*; *Yousef et al., 2015*). Factors activating Notch, growth differentiation factor 11 (GDF11), and tissue inhibitor of metalloproteinases 2 (TIMP-2), on the other hand, have been suggested to have rejuvenating effects (*Castellano et al., 2017*; *Conboy et al., 2005*; *Conboy and Rando, 2002*; *Mahmoudi et al., 2019*), although the role of GDF11 is still controversial (*Egerman et al., 2015*; *Egerman and Glass, 2019*; *Harper et al., 2016*; *Schafer et al., 2016*; *Sinha et al., 2014*). In addition, transcript levels of the prolongevity protein Klotho in circulating extracellular vesicles decrease with aging (*Sahu et al., 2021*).

The modulation of age affected signaling pathways, represents a promising alternative to supplying or inhibiting systemic factors for rejuvenation. However, the study of pathways that are responsive to systemic changes is complicated by the heterogeneous composition of tissues, which contain a multitude of cell types that communicate through secreted short-range signals. Systemic factors do not always act in a direct manner on tissue-resident cell populations but can instead trigger paracrine propagation and modulation of signals through accessory cell types in the niche. In the intestinal crypt, paneth cells are known to transmit systemic signals induced by caloric restriction to intestinal stem cells (*Yilmaz et al., 2012*). Moreover, fibro-adipogenic progenitors, an age-affected support cell population for skeletal muscle stem cells, are highly susceptible to systemic cytokines (*Biferali et al., 2019*; *Lukjanenko et al., 2019*). Factors in the aging circulation also alter accessory cell signaling in the neurogenic niche, which contributes to neural stem cell dysfunction (*Smith et al., 2018*). These examples illustrate that profiling of aging pathways that are directly affected by systemic long-range signaling factors requires an approach that allows to subtract the pervasive noise generated by the heterogeneous tissue environment.

Here, we present a method that allows for the encapsulation of homogeneous cell populations in diffusible hollow fiber capsules that can be transplanted subcutaneously to profile the effects of an aged systemic environment in the absence of short-range cellular interactions (*Figure 1a*). This approach makes it possible to expose cells from multiple species and different genetic backgrounds to heterotypic physiological environments to characterize gene-environment relationships. It provides a paradigm to disentangle direct effects mediated by systemic signals from cell intrinsic programming and input relayed by accessory cells in the tissue niche.

## Results

### Encapsulation of myogenic progenitors in PES hollow fiber membranes

Given their wide use in ultrafiltration applications, their oxidative, thermal, and hydrolytic stability, and their favorable mechanical properties, we chose polyethersulfone (PES) hollow fiber membrane (HFM) tubes for encapsulation of primary human (hskMPs) and C57BL/6J mouse derived skeletal

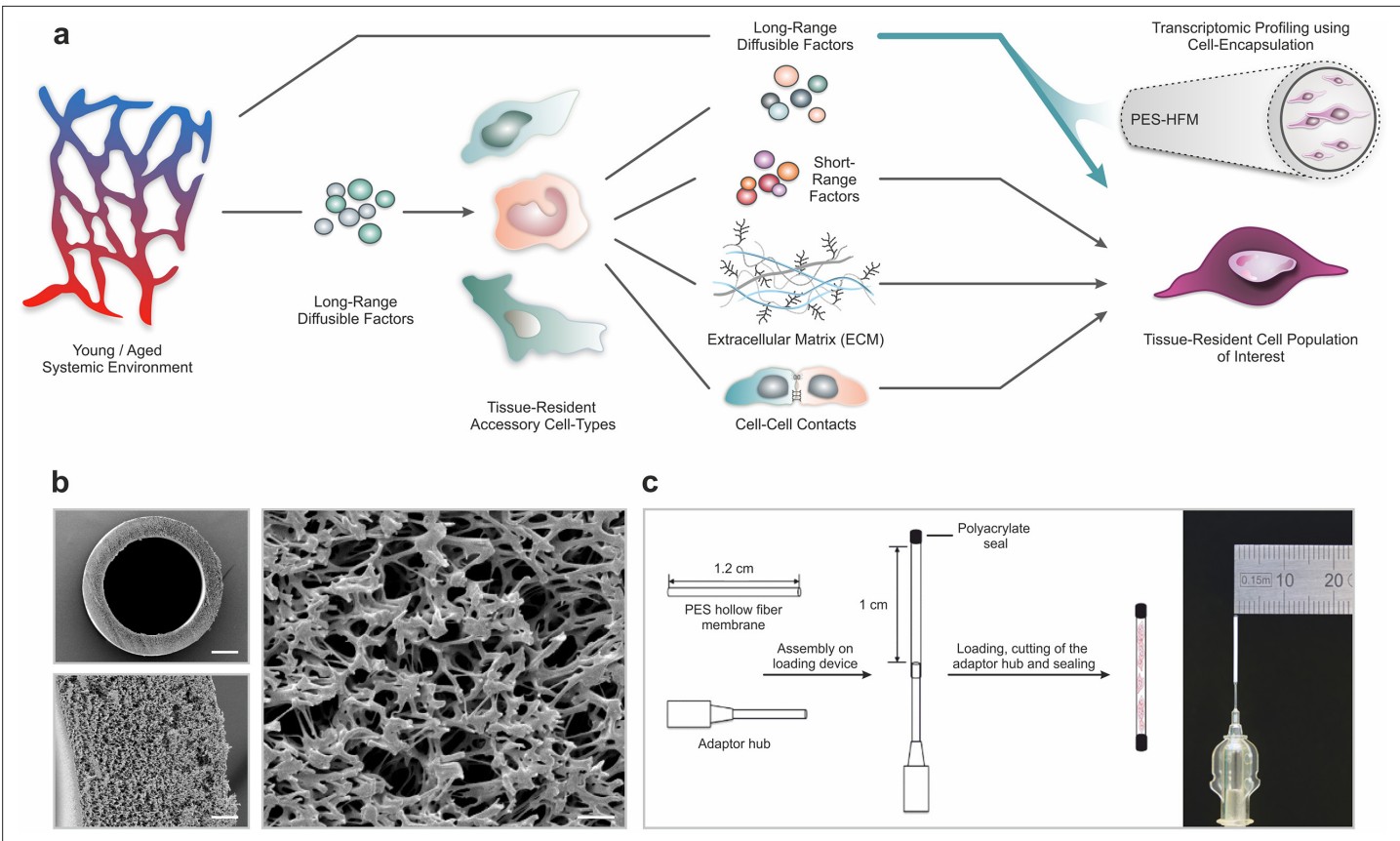

**Figure 1.** Polyethersulfone (PES) hollow fiber capsules for transcriptomic profiling of long-range systemic signals. (**a**) Scheme outlining interactions in the tissue niche in the context of the systemic circulation. Using cell encapsulation in PES hollow fiber membranes (PES-HFM), direct effects of long-range diffusible factors, including growth factors, nutrients, gases, and ions, on maintenance and differentiation of defined cell types of interest can be assessed independent of the influence of accessory cells presenting cell-cell adhesion receptors or secreting extracellular matrix and short-range signaling factors. (**b**) Scanning electron micrographs of a PES hollow fiber capsule. Scale bars: 200 μm (top left), 20 μm (bottom left), and 5 μm (right). (**c**) Schematic outlining the capsule loading procedure and photograph of a PES hollow fiber capsule mounted to the adaptor hub.

The online version of this article includes the following figure supplement(s) for figure 1:

**Figure supplement 1.** Description of the capsule loading and sealing platform and cellular characterization.

muscle progenitors (mskMPs) (**Zhao et al., 2013**). Importantly, PES fibers are biocompatible, have low immunogenicity, and become vascularized when subcutaneously implanted (**Hunter et al., 1999**). In order to maximize the number of encapsulated cells and, at the same time, allow for adequate oxygen diffusion throughout the capsule, we selected a tubular HFM with an outer diameter of 0.9 mm and an inner diameter of 0.7 mm (**Figure 1b**). The spongy-like, cross-linked architecture of PES-HFM allows for the diffusion of molecules up to 280 kDa freely through the membrane but prevents infiltration by cells. To provide a suitable adhesion matrix, cells were embedded in growth factor reduced Matrigel (**Kleinman et al., 1982**). Based on flow cytometric quantification of apoptosis, Matrigel resulted in cellular viability >90% following a 10-day culture period that was superior to other extracellular matrix substrates or hydrogel (**Figure 1—figure supplement 1a-c**). After mounting of the HFM to an adaptor hub, the exposed end and the external hub interface were sealed using a biocompatible photopolymerizing medical grade polyacrylate adhesive (**Figure 1c** and **Figure 1—figure supplement 1d**). Sealing of devices was verified using a submersion air pressure decay test. Trypsinized hskMPs or mskMPs mixed with Matrigel were injected into the capsule trough the adaptor hub and the loaded capsule was transferred onto a 3D printed autoclavable USP Class VI plastic cutting and sealing platform (**Figure 1—figure supplement 1e**,f). Following cutting of the adaptor hub, the loaded capsule remained protected in the UV impermeable plastic device while the second adhesive seal was photopolymerized.

## In vitro characterization of encapsulated myogenic progenitors

For validation of our experimental setup and to interrogate the behavior of encapsulated cells, we performed a series of in vitro experiments. Capsules loaded with hskMPs or mskMPs were placed into culture dishes, immersed in growth media, and were maintained on a horizontal shaker platform in a tissue culture incubator. Terminal deoxynucleotidyl transferase dUTP nick end labeling (TUNEL) staining of cryosections from capsules that were kept in culture for 10 days revealed that over 90% of the cells remained viable (*Figure 2a, b*). Indicative of proper oxygen and nutrient supply, cell density increased slightly in capsules in culture (*Figure 2c, d*). Following 10 days in culture, hskMPs and mskMPs were distributed across the entire diameter of the capsules (*Figure 2e, f*). Over the 10-day encapsulation time course, hskMPs and mskMPs showed a mild ≤37% reduction in the number of cells positive for the myogenic marker Pax7, while ≥80% of the cells remained positive for MyoD (*Figure 2g–l*). Indicating that the capsule 3D context favors the maintenance of myogenic markers, hskMPs in classic 2D culture downregulated Pax7 and MyoD expression by 62% and MyoD by 37% over the same 10-day period (*Figure 1—figure supplement 1g*). No apparent difference in the distribution of mitochondria was observed when hskMPs or mskMPs in 2D culture were compared to encapsulated cells (*Figure 2—figure supplement 1a–d*). When compared to cells in 2D culture, 3D exposure to Matrigel in the capsules led to a higher cytoskeletal complexity with an 123% increase in filopodia in mskMPs (*Figure 2—figure supplement 1a–e*). Growth factor deprivation over a 4-day differentiation period induced a robust ≥80% reduction of Pax7 in hskMPs and mskMPs, reduced MyoD by 43% in human cells, and, in spite of spatial restraints due to 3D embedding, increased the terminal differentiation marker myosin heavy chain (MHC) by ≥170% in both cell types (*Figure 2m-s*). After encapsulation, proliferative cells could be recovered from the capsules by enzymatic liberation (*Figure 3—figure supplement 1a*). Collectively, these observations demonstrate that encapsulated hskMPs and mskMPs remain viable and proliferative, retain the expression of myogenic markers, and are capable to respond to pro-differentiative signals.

## In vivo characterization of encapsulated myogenic progenitors

We next established the capsule implantation and recovery procedure in adult male C57BL/6J mice. Since it is easily accessible and in contact with the extensive subcutaneous vasculature, skeletal muscle, and bone, resembling the systemic environment myogenic progenitors are exposed to in situ, we chose the myofascia of the ribcage as an implantation site. The animals were anesthetized, and incisions were made on the back slightly posterior to each scapula. Three to four capsules containing hskMPs or mskMPs were introduced in varying locations into the subcutaneous fascia over the ribcage separated by 3–5 mm (*Figure 3a*). Capsule insertion was performed using a hypodermal plastic tube with a sliding metal plunger. In case of hskMPs, an immune reaction to secreted xenogeneic proteins was prevented by subcutaneous implantation of an osmotic pump supplying the immunosuppressant FK-506 on the flank contralateral to the capsules. After 10 days of in vivo insertion, the capsules showed external vascularization and minimal connective tissue build-up (*Figure 3b*). The porous capsule wall served as an effective barrier for host cells, and we did not observe any CD31 positive blood vessels that were able to penetrate the devices (*Figure 3—figure supplement 1b*). TUNEL apoptosis assays using cross sections from explanted capsules revealed that ≥80% of hskMPs and mskMPs remained viable (*Figure 3c, d*), and that Pax7, MyoD, and MHC were still expressed by the cells (*Figure 3e–m*). Out of a panel of 40 inflammatory markers, not a single factor was upregulated in serum from C57BL/6J mice that received capsules containing syngeneic cells in the absence of immunosuppression (*Figure 3—figure supplement 2a* and *Supplementary file 1a*). Thus, PES hollow fiber capsules do not cause an apparent immunogenic response.

## In vivo profiling of systemic aging

To apply our systemic profiling protocol, we implanted young and aged C57BL/6J mice with capsules containing hskMPs and mskMPs. hskMPs take about 10 days to fully fuse into myotubes (*Cheng et al., 2014*). Thus, in order to account for eventual differentiative effects of the systemic environment on the encapsulated cells, we chose this time point for isolation and analysis. The RNA yield was sufficient for genome-wide transcriptomic profiling for both cell types (*Figure 4—figure supplement 1a, b*). Gene set enrichment analysis using the hallmark database revealed that targets of the Myc and E2F family of transcription factors that have previously been implicated in aging and senescence were induced in

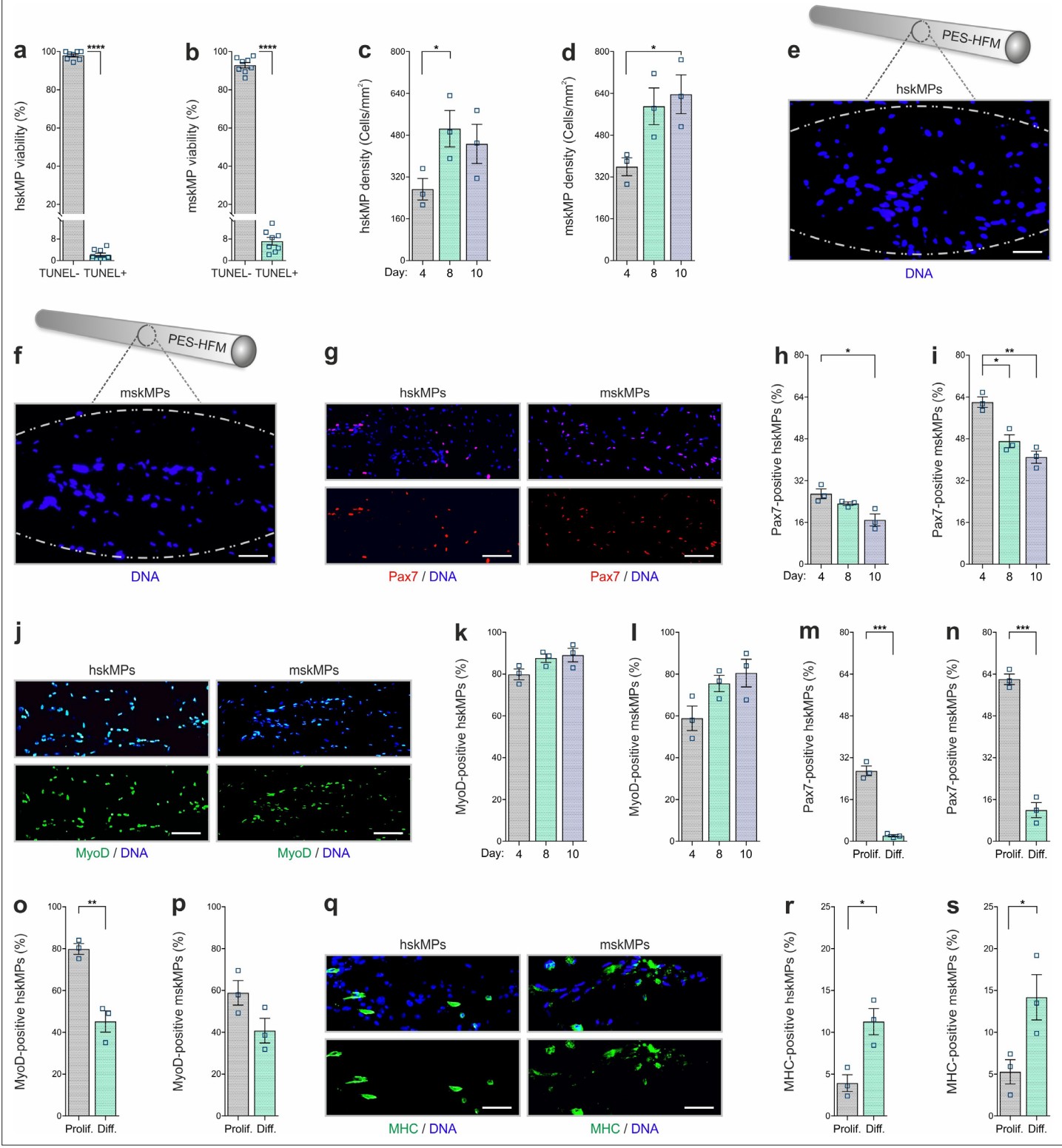

**Figure 2.** In vitro characterization of encapsulated myogenic progenitors. (**a**, **b**) Terminal deoxynucleotidyl transferase dUTP nick end labeling (TUNEL)-based quantification of apoptosis in encapsulated human (hskMPs) or mouse (mskMPs) skeletal muscle progenitors maintained in growth media for 10 days. (**c**, **d**) DNA staining-based quantification of hskMP or mskMP numbers in cross sections of capsules maintained for 4, 8, or 10 days in growth media. (**e**, **f**) Representative DNA stainings of cross sections from a capsule containing hskMPs or mskMPs maintained 10 days in growth media. Scale bars: 75 µm. (**g**) Representative Pax7 immunostainings of cross sections from capsules containing hskMPs or mskMPs maintained 10 days in growth

*Figure 2 continued on next page*

*Figure 2 continued*

media. Scale bars: 150 µm. (**h, i**) Quantification of Pax7 positive cells in capsules containing hskMPs or mskMPs maintained for 4, 8, or 10 days in growth media. (**j**) Representative MyoD immunostainings of cross sections from capsules containing hskMPs or mskMPs maintained 10 days in growth media. Scale bars: 150 µm. (**k, l**) Quantification of MyoD positive cells in capsules containing hskMPs or mskMPs maintained for 4, 8, or 10 days in growth media. (**m–p**) Quantification of Pax7 and MyoD in cross sections from capsules maintained for 4 days under proliferative (Prolif.) conditions in growth media or in differentiation (Diff.) media. (**q**) Representative myosin heavy chain (MHC) immunostainings of cross sections from capsules containing hskMPs or mskMPs maintained for 4 days in differentiation media. Scale bars: 75 µm. (**r, s**) Quantification of MHC positive cells in capsules containing hskMPs or mskMPs maintained for 4 days in growth media compared to differentiation media. All graphs represent means ± s.e.m. n ≥ 3 cross sections from different capsules were quantified for each experiment and time point. ****$p < 0.0001$, ***$p < 0.001$, **$p < 0.01$, *$p < 0.05$. Two-way comparisons were made with a Student's t-test and multiple comparisons by one-way ANOVA followed by Bonferroni post hoc test.

The online version of this article includes the following figure supplement(s) for figure 2:

**Figure supplement 1.** Comparison of encapsulated cells to 2D culture.

---

both hskMPs and mskMPs exposed to an aged systemic environment (*Dimri et al., 2000*; *Hofmann et al., 2015*; *Shavlakadze et al., 2018*; *Figure 4a–c* and *Supplementary file 1b–e*). To confirm these findings independently, we performed semi-quantitative PCR using mRNA isolated from capsules containing hskMPs after 10 days of implantation in young and aged mice. This experiment confirmed age-mediated a 157% increase of the Myc target *small nuclear ribonucleoprotein polypeptide A'* (*Snrpa1*) and a 92% increase of the E2F target *transferrin receptor* (*Tfrc*) that were part of the respective gene sets upregulated in encapsulated cells in old mice (*Figure 4—figure supplement 1c, d*).

Myc is induced by mitogens and inflammatory processes (*Frank et al., 2001*; *Liu et al., 2015*). However, aging is known to downregulate systemic sex hormones and the somatotroph axis leading to decreased levels of critical growth factors and mitogens such as IGF-1 and growth hormone (*Garcia et al., 2000*). Thus, in order to determine whether Myc induction goes along with elevated systemic proinflammatory markers, we profiled serum from aged C57BL/6J mice and detected increased levels of B lymphocyte chemoattractant (BLC, CXCL13), intercellular adhesion molecule-1 (ICAM-1, CD54), leptin, monokine induced by gamma (MIG, CXCL9), and TIMP-1 when compared to the young condition (*Figure 4—figure supplement 2a* and *Supplementary file 1f*). Interestingly, immunostaining for Ki-67 showed that increased levels of Myc signaling in the aged condition did not lead to a higher rate of proliferation in encapsulated cells (*Figure 4—figure supplement 3a, b*). However, we observed a 39% increase in β-galactosidase positive senescent cells in capsules explanted from aged mice (*Figure 4—figure supplement 3c, d*).

Further gene set analysis revealed that the categories myogenesis, epithelial–mesenchymal transition (EMT), interleukin, interferon, and p53 signaling were found to be suppressed in hskMPs and mskMPs by an aged systemic environment (*Figure 4d* and *Supplementary file 1g, h*). In agreement with an anti-myogenic effect in the aged circulation, we observed that encapsulated hskMPs explanted from old mice showed a 51% lower fusion index than in the young condition (*Figure 4—figure supplement 3e, f*).

To determine which features of aging are directly controlled by long-range secreted factors in the systemic circulation as opposed to signals transduced by the physiological tissue niche, we compared our dataset to freshly isolated myogenic progenitors from young and aged C57BL/6J mice (*Figure 4e* and *Supplementary file 1i, j*). Next to many exclusively niche controlled processes, we observed an overlap with respect to an upregulation of Myc and E2F targets, as well as a downregulation of myogenesis and EMT. Collectively, these results demonstrate that aging is associated with an increased abundance of systemic inflammatory molecules that correlates with higher activity of Myc and E2F transcription factors, cellular senescence, and a reduced differentiation potential of myogenic progenitors.

## Exposure to aged serum only captures a fraction of systemically affected pathways

We next set out to determine whether the systemic aging signature we observed using encapsulated cells in vivo can be reproduced using a simple cell culture paradigm. To this end we exposed hskMPs in 2D culture for 4 and 10 days to young and aged human serum, isolated RNA, and performed and genome-wide transcriptomic profiling. Gene set enrichment analysis revealed that aged human serum led to a weak induction of pathways at day 4, while more gene categories and a partial overlap with

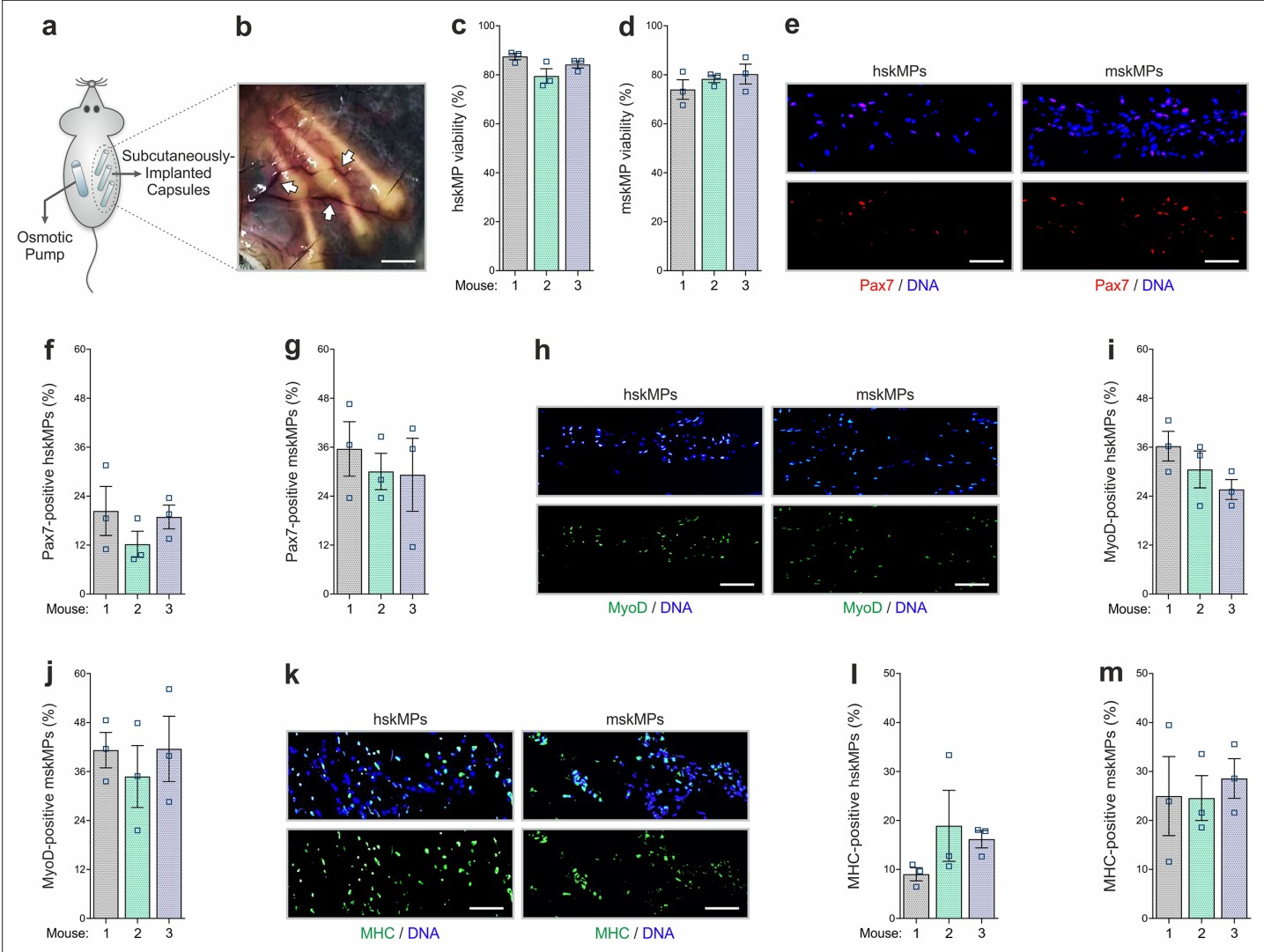

**Figure 3.** In vivo characterization of encapsulated myogenic progenitors. (**a**) Schematic of the capsule implantation strategy in mice. In case of hskMPs, an osmotic pump supplying immunosuppressant was implanted at the contralateral flank. (**b**) Photograph of the capsules in the connective tissue under the skin of mice 10 days after implantation. Arrows are pointing at blood vessels in proximity of the capsules. Scale bar: 0.2 cm. (**c**, **d**) Quantification of TUNEL negative hskMPs or mskMPs after 10 days of encapsulation in vivo. (**e**) Representative Pax7 immunostainings of cross sections from capsules containing hskMPs or mskMPs after 10 days in vivo. (**f**, **g**) Quantification of Pax7 positive cells in capsules containing hskMPs or mskMPs after 10 days in vivo. (**h**) Representative MyoD immunostainings of cross sections from capsules containing hskMPs or mskMPs after 10 days in vivo. (**i**, **j**) Quantification of MyoD positive cells in capsules containing hskMPs or mskMPs after 10 days in vivo. (**k**) Representative MHC immunostainings of cross sections from capsules containing hskMPs or mskMPs after 10 days in vivo. (**l**, **m**) Quantification of MHC positive cells in capsules containing hskMPs or mskMPs after 10 days in vivo. (**c**, **d**, **f**, **g**, **i**, **j**, **l**, **m**) Cross sections of capsules explanted from n = 3 mice were analyzed for each experiment. Graphs represent means ± s.e.m. (**e**, **h**, **k**) Scale bars: 150 μm. Comparisons were made by one-way ANOVA followed by Bonferroni post hoc test.

The online version of this article includes the following figure supplement(s) for figure 3:

**Figure supplement 1.** Enzymatically liberated encapsulated cells and staining for infiltration.

**Figure supplement 2.** Inflammatory profiling after capsule implantation.

the profile obtained from encapsulated myogenic progenitors in old mice was observed at day 10 (*Figure 4—figure supplement 4a* and *Supplementary file 1k, l*). In particular, the top three pathways, Myc targets, oxidative phosphorylation, and E2F targets, were similarly induced by aged human serum in vitro and in encapsulated myogenic progenitors in old mice. Counterintuitively, the category myogenesis was upregulated by aged serum in vitro after both 4 and 10 days of exposure. Moreover, other gene set categories such as fatty acid metabolism that were induced in both encapsulated cells

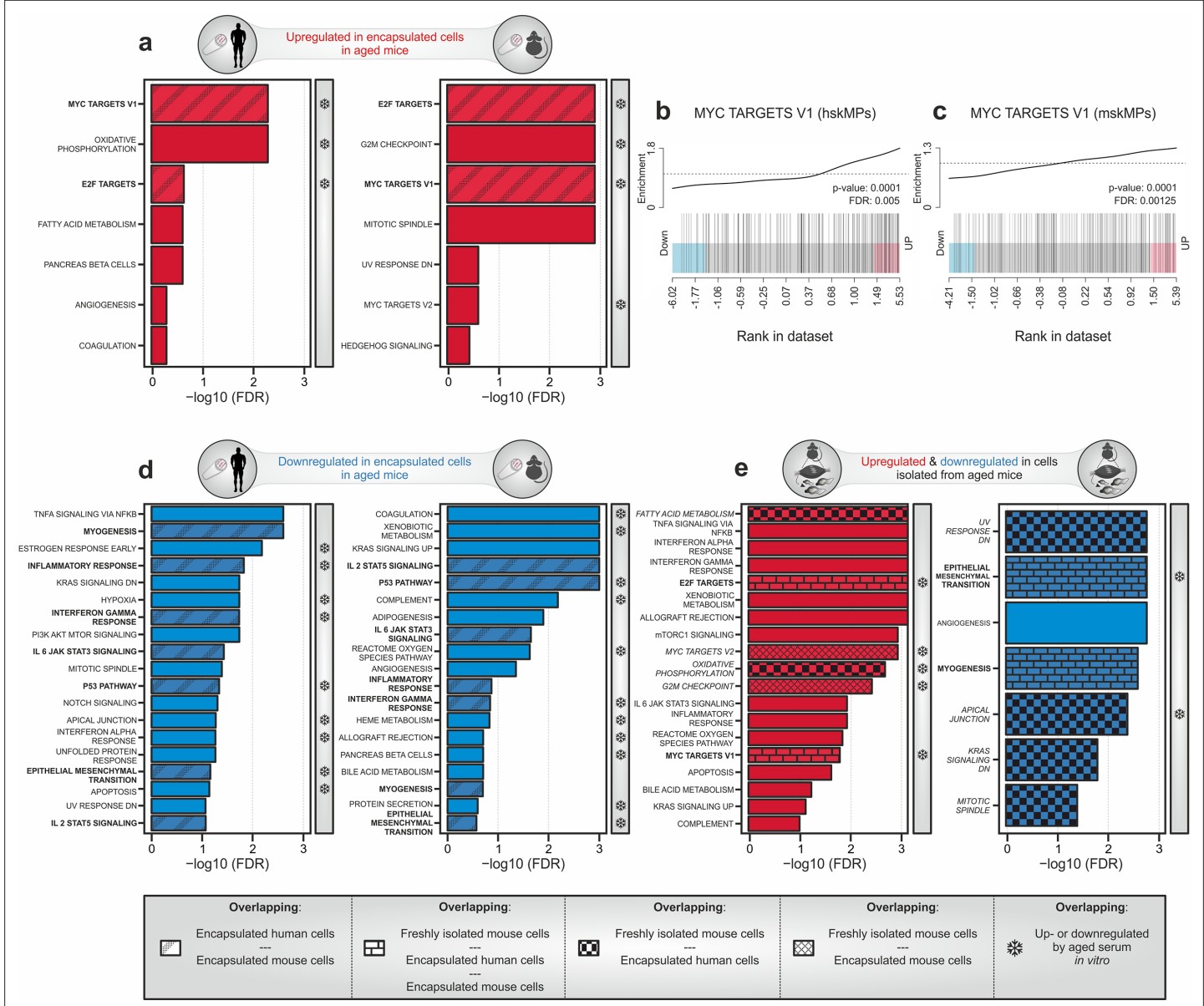

**Figure 4.** Transcriptomic profiling of systemic aging. (**a**) Gene set enrichment analysis (GSEA) of age-induced genes in encapsulated hskMPs or mskMPs compared to the young control after 10 days in vivo. Shaded bars represent gene sets overlapping between human and mouse cells. Snowflakes indicate gene sets that are also upregulated by aged serum in vitro (***Figure 4—figure supplement 4a***). (**b**, **c**) GSEA barcode plots depicting enrichment of Myc targets with age in encapsulated hskMPs or mskMPs compared to the young control after 10 days in vivo. (**d**) GSEA of genes downregulated with age in encapsulated hskMPs or mskMPs compared to the young control after 10 days in vivo. Shaded bars represent gene sets overlapping between human and mouse cells. Snowflakes indicate gene sets that are also downregulated by aged serum in vitro (***Figure 4—figure supplement 4b***). (**e**) GSEA of genes up- or downregulated with age in freshly isolated niche-resident skeletal muscle stem cells. Shaded bars represent gene sets overlapping with encapsulated hskMPs or mskMPs in young and aged mice. Snowflakes indicate gene sets that are also up- or downregulated by aged serum in vitro (***Figure 4—figure supplement 4a, b***). Data from n = 8 young or aged mice. (**a–e**) Samples and data are derived from capsules of n ≥ 5 mice in each age group. False discovery rate (FDR) = Adjusted p-value using Benjamini-Hochberg procedure.

The online version of this article includes the following figure supplement(s) for figure 4:

**Figure supplement 1.** RNA yield from capsules and validation of Myc and E2F target genes.

**Figure supplement 2.** Inflammatory profiling in young and aged mice.

**Figure supplement 3.** Proliferation, senescence, and differentiation of encapsulated cells after implantation in young and aged mice.

**Figure supplement 4.** Transcriptomic profiling of cells exposed to young and aged serum in 2D culture.

**Figure supplement 5.** Proliferation, senescence, and differentiation of cells after exposure to young and aged serum in 2D culture.

in old mice and in aged freshly isolated myogenic progenitors were not affected by aged serum in vitro.

After both 4 or 10 days of exposure of hskMPs to aged human serum in vitro, we observed a robust response in downregulated gene sets (*Figure 4—figure supplement 4b* and *Supplementary file 1m, n*). However, in particular, the top downregulated gene sets showed poor overlap with the profile obtained from encapsulated hskMPs or mskMPs in aged mice or in freshly isolated old myogenic progenitors. Moreover, myogenesis, which was consistently downregulated in both encapsulated cells and in freshly isolated cells in the aged condition, was not repressed by aged serum in vitro.

In contrast to our observations using encapsulated cells, we observed that aged human serum in vitro did not increase the abundance of senescence-associated β-galactosidase positive cells or affect the fusion index of the cells (*Figure 4—figure supplement 5a–f*). Overall, these results demonstrate that the effects of systemic aging observed in encapsulated myogenic progenitors in old mice can only partially be reproduced in vitro.

We conclude that cell encapsulation allows to capture the transcriptional effects of systemic aging on myogenic progenitors at unprecedented resolution. Long-range signals in the aged circulation lead to a deregulation of a broad spectrum of pathways, including an activation of Myc and E2F transcription factors, as well as an induction of anti-myogenic and senescence-related processes.

## Discussion

Our study demonstrates that encapsulation of syn- and xenogeneic cells allows for systemic transcriptional profiling independent of short-range heterogeneous cellular interactions. The effects of aged serum on cells in 2D culture only partially overlapped with our observations in encapsulated cells in young and old mice. This result supports the notion that certain factors in the systemic circulation are unstable in vitro or have a very short half-life. Compared to other types of capsules that have been used in mice, for instance planar macroencapsulation devices (*Lathuilière et al., 2014*), hollow fiber capsules are miniaturized to a diameter of 0.7 mm and a length of 1 cm that can be varied according to need. PES is one of the most frequently used polymers in medical applications and, due to its low immunogenicity, has been studied extensively in the context of artificial organs and medical devices used for blood purification in humans (*Samtleben et al., 2003*; *Tullis et al., 2002*; *Werner et al., 1995*; *Zhao et al., 2001*). Underlining its biocompatibility, we observed that subcutaneous implantation of PES capsules did not induce systemic inflammatory markers.

The myofascia of the ribcage is extensively vascularized and in proximity to bone and skeletal muscle. Thus, resembling the endogenous niche environment of myogenic cells, it was well suitable as an implantation site for our study. However, for investigation of other cell types that require implantation in different tissues or organs, the hollow fiber capsule format may not be ideal and may have to be adapted. Further miniaturization of the capsules would likely yield insufficient mRNA for bulk transcriptomics and either extensive amplification or single-cell sequencing would be required for downstream processing. Given, the broad interest of the aging field in using skeletal muscle stem cells as a model system for tissue maintenance and repair (*Drew, 2018*; *Evano and Tajbakhsh, 2018*; *Gopinath and Rando, 2008*), we selected myogenic progenitors for our profiling experiments. We observed that after viable proliferative cells can be enzymatically liberated from the capsules. Thus, our method is not limited to bulk transcriptomic profiling, but could also be modified to allow for single-cell sequencing or for the in vitro study of long-lasting intrinsic adaptations induced by exposure to an aged systemic environment.

Worldwide, the population of aged individuals has grown to an unprecedented size, and by 2050 more than 430 million people will be over the age of 80 (*Drew, 2018*). The characterization of bona fide aging signatures in defined cell populations independent of the heterogeneous niche context could potentially allow for the identification of novel therapeutic targets for the systemic treatment of age-associated cellular dysfunction in frail individuals. Importantly, cell encapsulation allows to filter out the dominant noise of the aged tissue that is in direct contact with the cell population of interest. Only long-range signaling factors are able to diffuse over the capsule membrane, which allows to read out effects of these molecules independent of extracellular matrix, heterologous cell-cell contacts, and short-range paracrine growth factors secreted by accessory cells in the tissue. Therefore, in contrast to profiling of cells directly extracted from aged niches or following parabiotic pairing, encapsulation allows to obtain a pure and unbiased transcriptional signature of the systemic environment in defined

cell types of choice and makes this amenable to cellular signaling across different species. As such, encapsulation is not an alternative to studying cells in their endogenous niche, but a complementary approach that allows to dissect specific questions on systemic versus tissue-mediated interactions.

Using gene set enrichment analysis, we observed that the aged systemic environment induces senescence, is anti-myogenic, and activates the Myc and E2F family of transcription factors in both mouse and human myogenic progenitors. E2F1, which interacts with the retinoblastoma tumor suppressor, has been shown to have a role in promoting cellular senescence (*Dimri et al., 2000*). Mice haploinsufficient for Myc exhibit an increased life span and are resistant to osteoporosis, cardiac fibrosis, and immunosenescence (*Hofmann et al., 2015*). Rapalogs, which inhibit the activity of mammalian target of rapamycin complex 1, increase life span and delay hallmarks of aging in many species. It has been shown that the rapalog RAD001, counter-regulates Myc in aged kidneys (*Shavlakadze et al., 2018*). Moreover, Myc is known to be induced by mitogens and inflammatory processes (*Frank et al., 2001*; *Liu et al., 2015*). In agreement with these observations, we detected significantly increased levels of systemic proinflammatory factors in aged mice. Interestingly, fundamental biological mechanisms, such as RNA processing and inflammation-related processes affected in our encapsulation study, were also changed in brain tissue of heterochronic parabionts (*Baruch et al., 2014*). Altogether, our study suggests that these cellular mechanisms are a consequence of systemic aging that occur independent of the influence of heterologous tissue-resident accessory cells. In contrast, comparison to myogenic cells directly isolated from skeletal muscle tissue indicates that processes such as apoptosis are imposed by the aged niche and are not directly affected by the systemic environment.

Importantly, our protocol is not limited to aging and might also allow to assess the impact of other multisystemic conditions on defined cell populations of interest. Moreover, in future applications, the combination of encapsulation technology with genetically engineered or induced pluripotent stem cell derived cells could allow to study systemic effects under controlled physiological conditions at an unprecedented level of detail.

## Materials and methods

### Cell culture

Human skeletal myoblasts (hskMPs, Lonza, CC-2580) isolated from donated human tissue of 20-year-old healthy Caucasians were used at passages 3–5 after obtaining permission for their use in research applications by the Cantonal Ethical Commission of Canton de Vaud (CER-VD). For in vitro expansion, the cells were maintained in human skeletal muscle myoblast growth medium (Zenbio, SKM-M) in human fibronectin coated dishes fibronectin (Corning, 356008) in a 37°C, 5% $CO_2$ incubator, and were passaged once confluency reached 50%. Primary mouse myoblasts (mskMPs) from 3-week-old C57Bl6/J (Charles River Canada, C57BL/6NCrl) mice were maintained in collagen I coated dishes (Sigma-Aldrich, C3867-1VL) in Ham's F10 media (Wisent, 318–051CL) containing 20% FBS (Wisent, 80450, lot 115714), 1% penicillin-streptomycin solution (Wisent, 450–201-EL) and 2.5 ng/mL bFGF (VWR, 10821-962) in a 37°C, 5% $CO_2$ incubator at a confluence under 80%. For transcriptomic profiling of hskMPs in vitro, the cells were seeded into human fibronectin coated dishes (Corning, 356008) in human skeletal muscle myoblast growth medium (Zenbio, SKM-M). After 8 hr, the cells were washed and maintained in human skeletal muscle myoblast growth medium (Zenbio, SKM-M) containing 9% of human serum from three different 19- to 20-year-old (young) or three different 60- to 64-year-old (aged) healthy Caucasian donors (HumanCells Biosciences, FP-006-C200) after obtaining permission for their use in research applications by informed consent and legal authorization for 4 or 10 days. The cells were maintained under humidified conditions at 37°C in a 5% $CO_2$ incubator and medium was changed every other day.

### Embedding of cells

Growth factor reduced Matrigel (Corning, 354230) and MaxGel ECM (Sigma-Aldrich, E0282) were thawed at 4°C and pipette tips were chilled at –20°C before starting the experiment. All material was kept on ice or at 4°C during the procedure. All mixing steps were carried out with caution to avoid generating bubbles. Cells were harvested with trypsin, numbers were quantified using a cell counter (Vi-CELL, AT39233), and the sample was centrifuged. The pellet was resuspended in ice-cold medium at a concentration of 20 k cells/µl and kept on ice. The cell mixture was mixed with 1 volume Matrigel

or MaxGel by gentle pipetting, giving rise to a final concentration of 10 k cells/μl. Hydrogels (Sigma-Aldrich, TrueGel3D Hydrogel Kit, TRUE7) of ~10 kPa were prepared by thawing of the SLO-DEXTRAN solution, TrueGel3D buffer at room temperature. SLO-DEXTRAN solution and TrueGel3D buffer were then mixed with the cells as described above for Matrigel and MaxGel. For hydrogel polymerization peptide-based cross-linker (Sigma-Aldrich, TRUECD) was added to the cell suspension mix.

## Device mounting

PES HFM (AKZO NOBEL) were cut into pieces of 1.2 cm (*Figure 1b, c*). The adaptor hub (*Supplementary file 1d*) was produced by assembling a plastic loading head (Neurotech Pharmaceuticals) with a piece of PEBAX single lumen tubing (Medical Extrusion Technologies). The piece of HFM was then connected to the adaptor hub and sealed at the external interface and the exposed end using a biocompatible photopolymerizing medical grade adhesive (Loctite, Henkel, L37DAI9124). Polymerization was induced using a BlueWave LED Prime UVA high-intensity spot-curing system (Dymax, 40322) emitting two 5 s pulses of UV. The remaining empty lumen (1 cm) of the capsule holds a volume of 4 μl. Sealing of each device was verified using an air-leak test. While immerged in sterile double distilled water, filtered air was injected at a pressure of 17.58 hPa (2.5 psi) for 5 s. Devices showing pressure decay greater than 100 Pa over 5 s were discarded. Assembled devices were sterilized with ethylene oxide gas before further use.

## Cell encapsulation

Ten μl of the cell-Matrigel mixture was loaded into each capsule using a Hamilton gastight syringe (50 μl) through the adaptor hub. Once the injected volume exceeded the inner volume of the device, a fraction of the total volume was ultrafiltrating through the porous membrane. The loaded capsule was transferred onto the 3D printed autoclavable USP Class VI plastic cutting and sealing platform (*Supplementary file 1e, f*), cut with a razor blade, and sealed with the medical grade adhesive while left protected in the UV blocking plastic device. The capsule was then transferred to pre-warmed media and maintained on a shaker (80 rpm) in a 37°C, 5% $CO_2$ incubator. Media was changed every day. For in vivo studies, freshly loaded capsules were maintained in the incubator overnight before implantation. To re-isolate live cells from capsules, they were cut with a razor blade on both ends. A Hamilton gastight syringe was then used to perfuse with StemPro accutase (Thermo Fisher Scientific, A1110501).

## Surgery

Capsule implantation experiments were performed using 6-week to 22-month-old male C57BL/6J mice (Janvier France, C57BL/6JRj) in accordance with the Swiss regulation on animal experimentation and the European Community Council directive (86/609/EEC) for the care and use of laboratory animals. Experiments were approved by the Vaud cantonal authorities under license VD3085, and by the Animal Care and Ethics Committee of the Spanish National Cardiovascular Research Center (CNIC) and regional authorities. Mice had access to water and food ad libitum at all time. Animals were randomized by body weight within experimental groups. Before surgery, mice were anesthetized using isoflurane and lidocaine was applied onto the shaved skin. Capsules were implanted through a small incision on the back, slightly posterior to the scapulae. Separated by 3–5 mm, three to four capsules were inserted through the incision into the subcutaneous fascia over the ribcage using hypodermic venflon plastic tube (BD) sliding over a metal plunger. The metal plunger held the capsule in place while the plastic applicator tube was withdrawn over it. For encapsulated human cells, an osmotic minipump (Alzet 1002, Charles River) supplying 2.5 mg/kg/day of FK-506 in 70% ethanol (VWR Chemicals BDH, 153386F) was implanted through a second incision on the opposite side of the spine. Subsequently the incisions were closed using surgical staples. After surgery, mice were kept in single housing with daily surveillance and bodyweight measurement. Ten days after implantation, the mice were euthanized and the capsules were retrieved, washed in warm PBS, incubated in 37°C warm trypsin for 5 min, and washed again before processing. Mice that showed weight loss >15% or displayed signs of wound infection and inflammation were excluded from the study.

## Cryo-sample preparation

Gelatin solution was heated up to and kept at 39°C until completely melted. Capsules were placed on a layer of gelatin applied to a thin plastic mold. Subsequently, another layer of gelatin was applied to cover the capsules. The sample was then kept at 4°C for 5 min to ensure complete gelation. The sample was then snap-frozen in a liquid nitrogen chilled isopentane slurry for 1 min and transferred to dry ice.

## Stainings

Capsule cryosections were fixed with 4% PFA (Thermo Fisher Scientific, 28908). Fixed samples were washed with PBS and permeabilized in 0.1% Triton X-100 (Sigma-Aldrich, T8787) for 15 min at room temperature. The sections were then blocked with 4% IgG-free BSA (Jackson ImmunoResearch, 001000162) for 1 hr at room temperature. Samples were incubated with primary antibody at 4°C overnight or for 2 hr at room temperature in blocking buffer. After washing, the sections were incubated with the corresponding secondary antibodies and 40, 6-diamidino-2-phenylindole (Thermo Fisher Scientific, D1306) for 45 min at room temperature. After further washing, the slide was dried and mounted (ProLong Diamond Antifade Mountant, P36965). Imaging was carried out using a DMI6000 inverted microscope (Leica, DM14000B) or VS120 slide scanner (Olympus, EVK-L100-042FL). Primary antibodies were anti-Pax7 (DHSB, 528428), anti-MyoD antibody (C-20) (Santa Cruz Biotechnology, sc-304), anti-MHC (Merck Millipore, A4.1025), anti-CD31 (Abcam, ab32457), Ki-67 (ab833, Abcam), and anti-β-galactosidase (Abcam, ab9361). Hoechst (B2261, Sigma-Aldrich) was used to stain DNA. Mitochondria were labeled using MitoTracker Red CMXRos (Thermo Fisher Scientific, M7512) and F-actin staining was performed using CytoPainter-Phalloidin-iFluor 488 reagent (Abcam, 176753). TUNEL staining was performed using the In Situ Cell Death Detection Kit (Roche, 11684795910) according to the manufacturer's instructions.

## Fluorescence-activated cell sorting

Cells were isolated after 4 and 10 days of embedding into Matrigel, MaxGel, and hydrogel using the TrypLE express enzyme (Thermo Fisher Scientific, 12605010) and StemPro accutase (Thermo Fisher Scientific, A1110501). Cell viability was determined using a LSRFortessa SORP FACS analyzer (BD Biosciences, H647800N0001). CytoCalcein (Pacific Blue) and Apopxin (FITC) (Abcam, ab176749) were used as indicators for live and apoptotic cells. Data was recorded with the BD FACSDiva software version 8.0.2. All data were subsequently analyzed with FCS Express Flow Cytometry version 6.06.0014 (De Novo Software, 4193).

## Mouse inflammatory cytokines

Serum cytokines were quantified using the enzyme-linked immunosorbent assay (ELISA)-based Quantibody Mouse Inflammation Array Q1 Kit (Raybiotech, QAM-INF-1–1). Following incubation of the cytokine-specific immobilized antibodies with serum and standard cytokines, a biotinylated antibody cocktail recognizing the different bound cytokines was added. For detection, Cy3-labeled streptavidin was added, and fluorescence was quantified using the InnoScan 710 AL microarray scanner (Innopsys, Innoscan-710). Data was extracted and computed using MAPIX software (Innopsys, version 8.2.7) and Quantibody Q-Analyzer software (Raybiotech, QAM-INF-1-SW). Cytokine concentration in the samples was determined by comparing signals from unknown samples to the control cytokine standard curve.

## RNA extraction and transcriptomic analysis

Two to three devices from the same mouse were pooled and added to a Lysing Matrix D tube (MP Biomedicals, 116913500) on ice. After addition of 450 µl of Agencourt RNAdvance Tissue lysis buffer (Beckman Coulter, A32646) the capsules were homogenized using a FastPrep-24 (MP Biomedicals). RNA was extracted using the Agencourt RNAdvance Tissue Kit (Beckman Coulter, A32646) following the manufacturer's instructions. For encapsulated hskMPs, two rounds cRNA synthesis starting with 5 ng of total RNA were performed using the MessageAmp II aRNA amplification kit (Life Technologies, AM1751) and MessageAmp II-biotin enhanced aRNA amplification kit (Life Technologies, AM1791) according to the manufacturer's instructions. RNA and cRNA were quantified using the Quant-iT RiboGreen RNA Assay Kit (Invitrogen, 10207502) using a Spectramax M2 (Molecular

Devices, M2). RNA quality assessment was performed using a Bioanalyzer 2100 with RNA 6000 Pico Kit (Agilent Technologies, 5067-1513). cRNA quality assessment was done using a Fragment Analyzer-96 with the Standard Sensitivity RNA Analysis Kit (15-nt) (Advanced Analytical Technologies, DNF-471-0500). Hybridization of 750 ng of cRNA on Human HT-12 v4.0 Expression BeadChip (Illumina, BD-103–0604) was performed according to the manufacturer's instructions. Scanning of the microarrays was performed on Illumina HiScan (Illumina, SY-103–1001). No signal was observed on human-specific microarrays using similar amounts of RNA/cRNA from connective tissue isolated in the immediate periphery of the implants. For encapsulated mskMPs, 50 ng of total RNA was used to generate QuantSeq libraries using the QuantSeq-3' mRNA-Seq-Library Prep Kit FWD for Illumina (Lexogen, 15.384) following 20 cycles of PCR amplification. Libraries were quantified with the Quant-iT Picogreen (Invitrogen, 10545213) on a FilterMax F3 (Molecular Devices, F3). Size pattern was assessed with Fragment Analyzer-96 with the DNF-474-0500 High Sensitivity NGS Fragment Analysis Kit (Agilent Technologies, DNF-474–0500). Libraries (average size: 295 bp) were pooled at an equimolar ratio and clustered at a concentration of 9 pM on a single read sequencing flow cell; 65 cycles of sequencing were performed on an Illumina HiSeq 2500 (Illumina, SY-401-2501) in rapid mode using a 50 cycles SBS Kit (Illumina, GD-402-4002, FC-402-4022) according to the manufacturer's instructions. The generated data were demultiplexed using bcl2fastq v2.19. Reads were aligned to the mouse genome (GRCm38) using STAR (*Dobin et al., 2013*), and the number of reads mapped within genes was quantified by HTSeq (*Anders et al., 2015*). Samples had a sequencing depth between 6.9 and 14.6 million reads, of which between 4.9 and 10.6 million reads were uniquely mapped. Freshly isolated niche-resident cells were isolated and transcriptomically analyzed as previously described (*Lukjanenko et al., 2016*). Briefly, extracted RNA were subjected to 3' microarray analysis on Illumina MouseRef–8_V2 chips. Semi-quantitative PCR was performed using a LightCycler 480 (Roche Diagnostics) and the LightCycler DNA green master mix (Roche Molecular Systems, 05573092001). Taqman probes (Thermo Fisher Scientific) were *Tfrc* (Hs00951083_m1), *Snrpa1* (Hs00795392_mH), and *GAPDH* (Hs02758991_g1) as a housekeeper.

## 3D printing

Design of the capsule cutting-sealing platform was carried out using Solidworks software (Dassault Systèmes, SW PRO). Printing was done using the ProJet 3500 HDMax (3D systems) 3D printer. VisiJet M3 Crystal and VisiJet S300 (3D systems, 1.0000-M06 and 1.0000-M03) served as printing material and support material, respectively. Schematics are deposited on https://www.thingiverse.com/thing:4005301.

## Statistical analysis

After visual inspection and exclusion of microarrays presenting low signal (log2 median expression <6) or low variability (standard deviation <0.1), Illumina expression signals were quantile-normalized. We applied a nonspecific filter to discard probes with low average signal and retained 6969 Illumina probes whose mean expression was greater than the third quartile of expression of all probes. Genes (represented by probes) were tested for differential expression using the moderated t-statistic as implemented in LIMMA (*Smyth, 2004*). RNA-sequencing data were normalized by the trimmed mean of M-values method using the calcNormFactors function in edgeR (*Robinson et al., 2010*) after selecting genes with more than four counts per million in at least five samples. Differentially expressed genes were defined by fitting a quasi-likelihood negative binomial generalized log-linear model to count data using glmQLFTest function in edgeR. The mean-rank gene set enrichment (*Michaud et al., 2008*) procedure as implemented in LIMMA was applied to investigate pathway perturbations between gene profiles derived from encapsulated cells exposed to a young and aged environment using the hallmark and C5 (GO) gene set collections from MSigDB (*Liberzon et al., 2015*) version 6.2. All genome-wide statistical analyses were performed using R, version 3.3.3 (microarray data), version 3.5.3 (RNA-sequencing data) and Bioconductor libraries. For cytokine arrays, the mean fluorescence intensities of positive controls were utilized for normalization. Values below the limit of detection (LOD) were substituted by LOD/sqrt(2) while the values above the highest standard were replaced by the highest standards. The non-parametric statistical (two-sample Wilcoxon) tests were used for the analysis. For visualization data were log-transformed.

## Accession codes

The data discussed in this publication have been deposited in NCBI's Gene Expression Omnibus (GEO). GEO Series accession numbers are GSE111401, GSE81096, and GSE193665.

## Acknowledgements

We are grateful to Nagabhooshan Hegde, José Sanchez, and the Nestlé Institute of Health Science Musculo-Skeletal Health and Cell Biology departments and community for discussion and support. We thank Phoukham Phothirath and Oliver Rizzo of the preclinical investigations group of Nestlé Research for expert advice and support with in vivo experiments. OM, JNF, and CFB are supported by the Fondation Suisse de Recherche sur les Maladies Musculaires (FSRMM). CFB is supported by the Canadian Institutes of Health Research (CIHR, PJT-162442), the Natural Sciences and Engineering Research Council of Canada (NSERC, RGPIN-2017-05490), the Fonds de Recherche du Québec – Santé (FRQS, Dossiers 296357, 34813, and 36789), the ThéCell Network (supported by the FRQS), the Canadian Stem Cell Network, and a research chair of the Centre de Recherche Médicale de l'Université de Sherbrooke (CRMUS). PMC is supported by ERC-2016-AdG-741966, La Caixa-HEALTH-HR17-00040, MDA, AFM, MWRF, UPGRADE-H2020-825825, RTI2018-096068-B-I00, a María de Maeztu Unit of Excellence award to UPF (MDM-2014-0370), and a Severo Ochoa Center of Excellence award to the CNIC (SEV-2015-0505). XH is recipient of a Severo Ochoa FPI (SEV-2015-0505-17-1) predoctoral fellowship.

## Additional information

### Competing interests

Omid Mashinchian, Xiaotong Hong, Joris Michaud, Gregory Lefebvre, Christophe Boss, Filippo De Franceschi, Sylviane Metairon, Frederic Raymond, Patrick Descombes, Nicolas Bouche, Jerome N Feige, C Florian Bentzinger: Presently or previously employed by the Société des Produits Nestlé S.A., Switzerland. Eugenia Migliavacca: Presently or previously employed by the Société des Produits Nestlé; S.A., Switzerland. The other authors declare that no competing interests exist.

### Funding

| Funder | Grant reference number | Author |
|---|---|---|
| Canadian Institutes of Health Research | PJT-162442 | C Florian Bentzinger |
| National Science and Research Council of Canada | RGPIN-2017-05490 | C Florian Bentzinger |
| Fonds de Recherche du Québec - Santé | Dossier 296357 | C Florian Bentzinger |
| Fonds de Recherche du Québec - Santé | Dossiers 34813 and 36789 | C Florian Bentzinger |
| Centre de Recherche Médicale de l'Université de Sherbrooke | CRMUS Chair | C Florian Bentzinger |
| European Research Council | ERC-2016-AdG-741966 | Pura Muñoz-Cánoves |
| La Caixa Foundation | La Caixa-HEALTH-HR17-00040 | Pura Muñoz-Cánoves |
| Muscular Dystrophy Association | MDA | Pura Muñoz-Cánoves |
| H2020 | UPGRADE-H2020-825825 | Pura Muñoz-Cánoves |
| Programa Estatal de Investigacion | RTI2018-096068-B-I00 | Pura Muñoz-Cánoves |

| Funder | Grant reference number | Author |
|---|---|---|
| Association Française contre les Myopathies | AFM | Pura Muñoz-Cánoves |
| MWRF | MWRF | Pura Muñoz-Cánoves |
| Maria de Maeztu Unit of Excellence award to UPF | MDM-2014-0370 | Pura Muñoz-Cánoves |
| Severo Ochoa Center of Excellence award to the CNIC | SEV-2015-0505 | Pura Muñoz-Cánoves |
| Severo Ochoa FPI predoctoral fellowship | SEV-2015-0505-17-1 | Xiaotong Hong |
| Fondation Suisse de Recherche sur les Maladies Musculaires (FSRMM) | | Omid Mashinchian Jerome N Feige C Florian Bentzinger |
| ThéCell Network (supported by the FRQS) | | C Florian Bentzinger |
| Canadian Stem Cell Network | | C Florian Bentzinger |

The funders had no role in study design, data collection and interpretation, or the decision to submit the work for publication.

## Author contributions

Omid Mashinchian, Conceptualization, Data curation, Formal analysis, Investigation, Methodology, Writing – original draft, Writing – review and editing; Xiaotong Hong, Data curation, Formal analysis, Investigation, Methodology, Writing – original draft, Writing – review and editing; Joris Michaud, Filippo De Franceschi, Sylviane Metairon, Data curation, Formal analysis, Methodology; Eugenia Migliavacca, Gregory Lefebvre, Frederic Raymond, Data curation, Formal analysis; Christophe Boss, Conceptualization, Methodology, Supervision, Writing – review and editing; Emmeran Le Moal, Resources, Writing – original draft; Jasmin Collerette-Tremblay, Resources; Joan Isern, Data curation, Methodology; Patrick Descombes, Resources, Supervision; Nicolas Bouche, Conceptualization, Supervision; Pura Muñoz-Cánoves, Conceptualization, Funding acquisition, Resources, Supervision, Writing – original draft, Writing – review and editing; Jerome N Feige, Conceptualization, Data curation, Formal analysis, Funding acquisition, Methodology, Project administration, Supervision, Writing – original draft, Writing – review and editing; C Florian Bentzinger, Conceptualization, Data curation, Formal analysis, Funding acquisition, Investigation, Methodology, Project administration, Resources, Supervision, Writing – original draft, Writing – review and editing

## Author ORCIDs

Joan Isern (iD) http://orcid.org/0000-0002-1401-9779
Jerome N Feige (iD) http://orcid.org/0000-0002-4751-264X
C Florian Bentzinger (iD) http://orcid.org/0000-0003-0422-9622

## Ethics

This study was performed in accordance with the Swiss regulation on animal experimentation and the European Community Council directive (86/609/EEC) for the care and use of laboratory animals. Experiments were approved by the Vaud cantonal authorities under license VD3085, and by the Animal Care and Ethics Committee of the Spanish National Cardiovascular Research Center (CNIC) and regional authorities.

## Decision letter and Author response

Decision letter https://doi.org/10.7554/eLife.57393.sa1
Author response https://doi.org/10.7554/eLife.57393.sa2

# Additional files

## Supplementary files

• Supplementary file 1. Tables. (a) Multiplexed enzyme-linked immunosorbent assay (ELISA) quantification of inflammatory factors in serum of mice that underwent surgical implantation of polyethersulfone (PES) hollow fiber capsules compared to untreated controls 10 days after the procedure. n ≥ 7 mice. Each value represents averages in pg/ml from n = 4 technical replicates for each factor. TNF-Rl and MIP-1g were excluded from the analysis since signals were outside of the detection range. (b) Gene sets increased with age in encapsulated human skeletal muscle progenitors (hskMPs) compared to the young control after 10 days in vivo. Data are derived from capsules of n ≥ 5 mice in each age group. GST p-value (Pval) = Wilcoxon gene set test p-value. GST FDR = Adjusted p-value using the Benjamini-Hochberg procedure. GST FDR = Wilcoxon gene set test false discovery rate. (c) Gene sets increased with age in encapsulated mouse skeletal muscle progenitors (mskMPs) compared to the young control after 10 days in vivo. Data are derived from capsules of n ≥ 5 mice in each age group. GST Pval = Wilcoxon gene set test p-value. GST FDR = Adjusted p-value using the Benjamini-Hochberg procedure. (d) Significantly enriched Hallmark Myc V1 target genes when comparing encapsulated hskMPs in aged mice to the young control after 10 days in vivo. Data are derived from capsules of n ≥ 5 mice in each age group. Pval < 1%. (e) Significantly enriched Hallmark E2F target genes when comparing encapsulated hskMPs in aged mice to the young control after 10 days in vivo. Data are derived from capsules of n ≥ 5 mice in each age group. Pval < 1%. (f) Multiplexed ELISA array quantification of levels of inflammatory factors in plasma of mice young and aged mice. n ≥ 5 mice. Each value represents averages in pg/ml from n = 4 technical replicates for each factor. TNF-Rl and MIP-1g were excluded from the analysis since signals were outside of the detection range. (g) Gene sets decreased with age in encapsulated hskMPs compared to the young control after 10 days in vivo. Data are derived from capsules of n ≥ 5 mice in each age group. GST Pval = Wilcoxon gene set test p-value. GST FDR = Adjusted p-value using the Benjamini-Hochberg procedure. (h) Gene sets decreased with age in encapsulated mskMPs compared to the young control after 10 days in vivo. Data are derived from capsules of n ≥ 5 mice in each age group. GST Pval = Wilcoxon gene set test p-value. GST FDR = Adjusted p-value using the Benjamini-Hochberg procedure. (i) Gene sets increased with age in niche-resident freshly isolated mouse muscle stem cells compared to the young control. Data from n = 8 young or aged mice. GST Pval = Wilcoxon gene set test p-value. GST FDR = Adjusted p-value using the Benjamini-Hochberg procedure. (j) Gene sets decreased with age in niche-resident freshly isolated mouse muscle stem cells compared to the young control. Data from n = 3 young or aged mice. GST Pval = Wilcoxon gene set test p-value. GST FDR = Adjusted p-value using the Benjamini-Hochberg procedure. (k) Gene sets increased after 4 days in 2D hskMP culture exposed to aged human serum compared to the young human serum. GST Pval = Wilcoxon gene set test p-value. Data are derived from hskMPs exposed to human serum from n = 3 different young or aged donors. GST FDR = Adjusted p-value using the Benjamini-Hochberg procedure. (l) Gene sets increased after 10 days in 2D hskMP culture exposed to aged human serum compared to the young human serum. Data are derived from hskMPs exposed to human serum from n = 3 different young or aged donors. GST Pval = Wilcoxon gene set test p-value. GST FDR = Adjusted p-value using the Benjamini-Hochberg procedure. (m) Gene sets decreased after 4 days in 2D hskMP culture exposed to aged human serum compared to the young human serum. Data are derived from hskMPs exposed to human serum from n = 3 different young or aged donors. GST Pval = Wilcoxon gene set test p-value. GST FDR = Adjusted p-value using the Benjamini-Hochberg procedure. (n) Gene sets decreased after 10 days in 2D hskMP culture exposed to aged human serum compared to the young human serum. Data are derived from hskMPs exposed to human serum from n = 3 different young or aged donors. GST Pval = Wilcoxon gene set test p-value. GST FDR = Adjusted p-value using the Benjamini-Hochberg procedure.

• Transparent reporting form

## Data availability

The data discussed in this publication have been deposited in NCBI's Gene Expression Omnibus (GEO). GEO Series accession numbers are GSE111401, GSE81096 and GSE193665.

The following datasets were generated:

| Author(s) | Year | Dataset title | Dataset URL | Database and Identifier |
|---|---|---|---|---|
| Bentzinger CF, Mashinchian O, Hong X, Michaud J, Migliavacca E, Lefebvre G, Boss C, Franceschi FD, Le Moal E, Collerette-Tremblay J, Isern J, Metairon S, Raymond F, Descombes P, Bouche N, Muñoz-Cánoves P, Feige JN | 2022 | In Vivo Transcriptomic Profiling using Cell Encapsulation Identifies Effector Pathways of Systemic Aging | https://www.ncbi.nlm.nih.gov/geo/query/acc.cgi?acc=GSE111401 | NCBI Gene Expression Omnibus, GSE111401 |
| Bentzinger CF, Mashinchian O, Hong X, Michaud J, Migliavacca E, Lefebvre G, Boss C, Franceschi FD, Le Moal E, Collerette-Tremblay J, Isern J, Metairon S, Raymond F, Descombes P, Bouche N, Muñoz-Cánoves P, Feige JN | 2022 | In Vivo Transcriptomic Profiling using Cell Encapsulation Identifies Effector Pathways of Systemic Aging | https://www.ncbi.nlm.nih.gov/geo/query/acc.cgi?acc=GSE193665 | NCBI Gene Expression Omnibus, GSE193665 |

The following previously published dataset was used:

| Author(s) | Year | Dataset title | Dataset URL | Database and Identifier |
|---|---|---|---|---|
| Lukjanenko L, Migliavacca E, Karaz S, Metairon S, Raymond F, Bentzinger F, Feige JN | 2016 | Profiling of mouse muscle stem cells aging and activation upon skeletal muscle injury | https://www.ncbi.nlm.nih.gov/geo/query/acc.cgi?acc=GSE81096 | NCBI Gene Expression Omnibus, GSE81096 |

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
