## [Editor Report]

The manuscript includes in vivo studies where encapsulated myogenic progenitors are exposed to the systemic environment of young or aged mice. The authors provide a very important comparison of a novel approach to the use of young or aged serum in vitro, which is considered the current gold standard. The studies reported also provide evidence that the in vivo capsule-based method may constitute an alternative and possibly improved approach to the study of impact of environment-related changes on function of skeletal muscle progenitors.

---

## [Decision Letter]

**Decision letter after peer review:**

Thank you for submitting your article "In-Vivo transcriptomic profiling using cell encapsulation identifies Myc and E2F as effectors of systemic aging" for consideration by *eLife*. Your article has been reviewed by 3 peer reviewers, one of whom is a member of our Board of Reviewing Editors, and the evaluation has been overseen by Jessica Tyler as the Senior Editor. The reviewers have opted to remain anonymous.

The reviewers have discussed the reviews with one another and the Reviewing Editor has drafted this decision to help you prepare a revised submission.

Summary:

The manuscript by Mashinchian et al., describes an innovative approach to dissect the effects of an age-associated systemic environment on myogenic progenitors encapsulated in diffusible artificial capsules and engrafted into young and old mice. They describe a method whereby mouse and human myogenic progenitors are suspended in a hydrogel matrix which is then cultured within thin plastic capsules that are placed subcutaneously below the skin of young and aged mice for 10 days, followed by transcriptomic analyses of the cell contained within the capsule.

The novelty of this manuscript stems from the development of a method to dissect the systemic environmental effects of aging.

This idea is of high interest and deserves further development, as it would provide a simplified model to understand the impact of the milieu on cell type-specific phenotypes, limiting inputs from direct cell-cell interactions. Studies of cultured cells directly implanted in different physiological environments in vivo would be a powerful tool to inform in vivo and ex vivo experiments, including but not limited to parabiosis studies.

In its current state, however,

(a) the submission lacks evidence that the method uncovers targets that would have been missed using a simpler approach (i.e. treating the cells in 2D or 3D culture with yng vs aged sera), making it hard to conclude whether the in vivo transplantation method provides an advantage over conventional approaches;

(b) conclusions about validating aging associated targets first identified by others are based exclusively on transcriptomics analysis, with no further validation;

(c) and as this is a methods submission, substantial additional basic characterization and validation is needed to establish robustness and reliability of the approach, at least for myogenic progenitors and ideally for at least another relevant cell type.

Essential revisions:

Studies should include (but not be limited to) descriptive and quantitative 3-D immunocytochemistry for markers indicative of overall architecture, structure and integrity of cytoskeleton and organelles, and quantitative measures of general and cell type-specific function (e.g. mitochondrial function, others relevant to muscle progenitor function). The capsule approach should also be validated functionally with 'positive' controls where the systemic response is already defined and well characterized. Those studies should also define the interindividual variation for the responses 'read' by the inserts.

Vascularization and other potential changes in the capsules after exposure to the in vivo milieu should be characterized and demonstrated using adequate quantitative approaches, providing scale bars in images, quantitative analyses for outcomes, etc.

To appropriately evaluate the transcriptomic studies the RNA yields per insert need to be clearly shown, using units of mass per unit volume or mass per insert. If average RNA yields from capsules inserted in young animals are taken as "100%" then it is unclear where the variance in percent values for the 'young' group arises from.

Because the stated goal of the studies is to develop a general tool, the usefulness of the encapsulation system for the study of more than one cell type should be demonstrated.

The authors use the word 'niche' as equivalent to cell-cell contacts – however these are only one component of a biological niche, that also include the specific local non-cellular milieu (i.e. specific nutrient and ion composition and levels, as well as specific secretomes, among other factors).

The sites where inserts were implanted (e.g. spaces that interface different organ systems, or specific areas of organs such as brain) are actually specific 'niches' that are different from the general circulation. The text needs to be revised such that suggested interpretation of the data obtained using the cell encapsulation approach described needs to be revised throughout the text.

The authors state that they would profile proinflammatory markers to "determine whether Myc induction is due to elevated systemic proinflammatory markers". The relation is one of association, not causation, so this sentence and others where association is described as causation need to be corrected.

The authors need to address the following points:

a) Is the assay necessary: The current gold standard to investigate direct effects of yng vs aged humoral factors on a specific cell population is to treat the cell population in culture with sera and then evaluate downstream cellular and molecular changes. Therefore, to demonstrate the value proposition of moving to the proposed in vivo assay, it is critical to show that the method is sensitive enough to detect known biology, that it can identify new biology, and that the new biology could not be discovered using the current gold standard approach.

b) Is the assay predictive: At the moment, the evidence that Myc and E2F are induced in aged myogenic cells is based on transcriptomic studies, and the conclusion that this aberrant signaling is driven by inflammatory processes is based on correlation to sera profiling. Further, validation of Myc / E2F as elements of an aging signature is based on comparison of comparing the transcriptomic profiling of myogenic progenitors culture in the capsules to freshly isolated muscle stem cells and brain tissue. Additional modes of validation are warranted to demonstrate the relevance of Myc / E2F in muscle cells and the predictive nature of the method.

c) Is the assay optimized: The rationale for assay design criteria lack justification and therefore seem arbitrarily selected, which is unusual for a method paper. For example; (a) how was the analysis time-point selected to be Day 10? How does this choice relate to obtaining data that predicts aging factors that directly modulate muscle stem cells?; (b) was there a rationale in selecting a cell seeding density that is close to that used in skeletal muscle tissue engineering applications to create differentiated myotubes?; (c) with muscle stem cells as the desired comparator, were there desired criteria with regards to the differentiation status of the cells within the capsule?; (d) two reconstituted basement membrane (human and mouse) ECMs (~200 pascals) and TrueGel a non-adherent hydrogel (~10 kPa) were compared as scaffolds – what was the rationale for selecting these particular materials as it related to your experimental design?; (e) etc

While the authors performed an elegant analysis to demonstrate that the myogenic progenitors remain viable and maintain stem cell attributes in capsules, the differentiation assay is based on fuzzy MyHC immunofluorescence and the stained cells do not show multinucleated phenotype as expected. Can the authors comment on the reason their cells did not fuse, or alternatively provide higher resolution images showing multinucleated fibers? This is of particular importance given that they refer to this assay as "terminal differentiation"- however do not provide sufficient evidence of multinucleated muscle formation.

The authors need to validate using an additional assay aside from RNA-seq (i.e. RT-qPCR or immunofluorescence) the upregulation of select Myc or E2F targets in myogenic progenitors exposed to the aged mouse environment. On a similar note, they need to provide a full or at least partial list of these target genes.

Given the upregulation of targets of E2F it will be of interest to compare the expression level of canonical cell cycle regulators and senescent markers in myogenic progenitors exposed to old and young mouse environments. It is of further interest to assess if cells exposed to aged environment demonstrate less proliferation via EDU or Ki67 staining and upregulation of senescent markers.

The category "Myogenesis" is reported as one of the two primary downregulated clusters in myogenic cells exposed to an aged mouse environment. To corroborate this interesting observation, it will significantly strengthen the manuscript if the authors assess whether myogenic progenitors exposed to an old mouse environment exhibit reduced myogenic differentiation potential. This can be done via quantification of MyHC staining or analysis of myotube fusion index between cells exposed to the two conditions.

The RNA-Seq data analysis points to downregulation of inflammatory response in cells exposed to old vs. young systemic environment, however the authors also concomitantly report on elevated levels of systemic inflammation markers detected in blood of aged mice. Can the authors comment or discuss the discrepancy between these two opposing observations?

The comparison to a previously published dataset from a parabiosis trial is an underdeveloped off-shoot and the message the authors wish to convey is supported by thin data analysis. The authors need to expand their analysis or discussion on the significance of this observation or alternatively consider removing it from the manuscript.

---

## [Author Response]

Essential revisions:Studies should include (but not be limited to) descriptive and quantitative 3-D immunocytochemistry for markers indicative of overall architecture, structure and integrity of cytoskeleton and organelles, and quantitative measures of general and cell type-specific function (e.g. mitochondrial function, others relevant to muscle progenitor function).

We addressed cell type-specific function of encapsulated cells by staining for the muscle stem cell self-renewal marker Pax7, the myogenic proliferation/activation marker MyoD, and the differentiation marker myosin heavy chain (Figure 2g-s). As suggested, we also performed a staining of mitochondria and the actin cytoskeleton in encapsulated cells (Figure 2—figure supplement 1a-e). In addition, we now provide staining and quantification of the proliferation marker Ki-67, the senescence marker β-galactosidase, and the differentiation marker myosin heavy chain in encapsulated cells after in vivo exposure to a young and aged systemic environment, as well as after culture in aged human serum in vitro (Figure 4—figure supplement 3 and 5).

The capsule approach should also be validated functionally with 'positive' controls where the systemic response is already defined and well characterized. Those studies should also define the interindividual variation for the responses 'read' by the inserts.

To our knowledge we are first to report the use of cell encapsulation for transcriptomic profiling of systemic effects in vivo and we were not able to identify a suitable well-characterized positive control. Thus, we chose to sequence myogenic progenitors exposed to young and aged human serum in vitro as an additional control experiment and reference point (Figure 4—figure supplement 4a,b). This experiment revealed that some of the main pathways that were deregulated in encapsulated cells in aged mice are similarly affected by aged serum in vitro. Importantly, in spite of this overlap, we observed that certain aging pathways are differentially regulated between the in vivo encapsulation condition and the in vitro paradigm using serum, which supports the notion that some systemic factors have a very short half-life or are not stable under cell culture conditions and can only be detected in the flux of the circulation. In addition, we demonstrate that the expression profiles of freshly isolated muscle stem cells from old mice partially overlap with the profile we observed using encapsulated cells in vivo (Figure 4e, Supplementary file 1i,j). Thus, we compared our novel in vivo profiling method to multiple positive reference points and the data supports and strengthens our conclusion that cell encapsulation allows for efficient profiling of systemic aging.

To account for interindividual responses in quantifications, we always provide datapoints representing the different mice and capsules. For instance, each bar in the graphs in figure 3 represents a different mouse and the overlaid datapoints signify different capsules obtained from the respective animals. In all other in vivo experiments, except the bioinformatic analysis, the values of each capsule from different mice are overlaid as datapoints with the bars representing averages. For instance, for the quantifications in Figure 4—figure supplement 3, each datapoint overlaid with a given bar represents a capsule explanted from a different mouse. All bioinformatic analyses include statistical analyses testing for significant differences in gene expression across biological replicates (i.e. independent RNA from capsules obtained from different mice or cell culture replicates) after correction for multiple testing through the false discovery rate.

Vascularization and other potential changes in the capsules after exposure to the in vivo milieu should be characterized and demonstrated using adequate quantitative approaches, providing scale bars in images, quantitative analyses for outcomes, etc.

We provide scalebars in all representative images for each readout after exposure to the in vivo milieu, as well as quantitative analysis with statistics that clearly specifies biological replicates (please also see response above).

The molecular weight cut-off of the capsules is 280 kDa, which is considered too small for cells to pass through. Thus, we could not detect any CD31 positive endothelial cells that are indicative of host derived vascularization in the devices (Figure 3—figure supplement 1b). To address other potential changes in the capsules, we now provide staining and quantification of the proliferation marker Ki-67, the senescence marker β-galactosidase, and the differentiation marker myosin heavy chain in encapsulated cells after in vivo exposure to a young and aged systemic environment (Figure 4—figure supplement 3a-f).

To appropriately evaluate the transcriptomic studies the RNA yields per insert need to be clearly shown, using units of mass per unit volume or mass per insert. If average RNA yields from capsules inserted in young animals are taken as "100%" then it is unclear where the variance in percent values for the 'young' group arises from.

We now provide the RNA yield in absolute numbers (Figure 4—figure supplement 1a,b) and the methods describe that each capsule contained 100k cells .

Because the stated goal of the studies is to develop a general tool, the usefulness of the encapsulation system for the study of more than one cell type should be demonstrated.

The main goal of our paper is to demonstrate the feasibility of transcriptomic profiling of systemic aging using encapsulated myogenic progenitors and to provide an overview of its effects on gene expression in this cell type. Thus, we replicated the majority of our study with two different myogenic cell types: Primary mouse myogenic progenitors and primary human myogenic progenitors. We now also elaborate in the discussion of our manuscript regarding the potential usefulness of our method for other non-myogenic cell types and how the system may have to be adapted for other tissues and organs .

The authors use the word 'niche' as equivalent to cell-cell contacts – however these are only one component of a biological niche, that also include the specific local non-cellular milieu (i.e. specific nutrient and ion composition and levels, as well as specific secretomes, among other factors).

To clarify the meaning of "niche" accordingly we illustrated the different components (short- and long-range signaling factors, ECM, and cell-cell contacts) in figure 1a. Based on the reviewers comments, we now specify in the respective figure legend that "long-range diffusible factors" not only include growth factors, but also nutrients, gases, and ions. In addition, in the discussion we elaborate again regarding different niche components:

"cell encapsulation allows to filter out the dominant noise of the aged tissue that is in direct contact with the cell population of interest. Only long-range signaling factors are able to diffuse over the capsule membrane, which allows to read out effects of these molecules independent of extracellular matrix, heterologous cell-cell contacts, and short-range paracrine growth factors secreted by accessory cells in the tissue." .

The sites where inserts were implanted (e.g. spaces that interface different organ systems, or specific areas of organs such as brain) are actually specific 'niches' that are different from the general circulation. The text needs to be revised such that suggested interpretation of the data obtained using the cell encapsulation approach described needs to be revised throughout the text.

To address this important point we now mention our choice of the implantation site in the Results section:

"Since it is easily accessible and in contact with the extensive subcutaneous vasculature, skeletal muscle, and bone, resembling the systemic environment of tissue-resident myogenic progenitors, we chose the myofascia of the ribcage as an implantation site." .

In addition, we elaborate on this topic again in the discussion:

"The myofascia of the ribcage is extensively vascularized and in close proximity to bone and skeletal muscle. Thus, due to its resemblance to the niche environment of tissue resident myogenic progenitors, it was well suitable as an implantation site for our study. However, for investigation of other cell types that require implantation in different tissues or organs the hollow fiber capsule format may not be ideal and may have to be adapted. Further miniaturization of the capsules would likely yield insufficient mRNA for bulk-sequencing and either extensive amplification or single-cell sequencing would be required for downstream processing." .

The authors state that they would profile proinflammatory markers to "determine whether Myc induction is due to elevated systemic proinflammatory markers". The relation is one of association, not causation, so this sentence and others where association is described as causation need to be corrected.

We have modified the entire manuscript accordingly and made sure to distinguish between causation and association. The example the reviewer highlighted has been changed as follows:

"in order to determine whether Myc induction goes along with elevated systemic proinflammatory markers" .

The authors need to address the following points:a) Is the assay necessary: The current gold standard to investigate direct effects of yng vs aged humoral factors on a specific cell population is to treat the cell population in culture with sera and then evaluate downstream cellular and molecular changes. Therefore, to demonstrate the value proposition of moving to the proposed in vivo assay, it is critical to show that the method is sensitive enough to detect known biology, that it can identify new biology, and that the new biology could not be discovered using the current gold standard approach.

In-spite of this being an obvious and interesting experiment, we were not able to identify a public transcriptomics dataset comparing myogenic progenitors cultured in young and aged serum. Therefore, we designed a study in which we exposed hskMPs in 2D culture for four and ten days to young and aged human serum, isolated RNA, and performed and genome wide transcriptomic profiling. Gene set enrichment analysis revealed that aged human serum led to a weak induction of pathways at day four, while more gene categories and a partial overlap with the profile obtained from encapsulated myogenic progenitors in old mice was observed at day ten (Figure 4—figure supplement 4a and Supplementary file 1k,l). In particular, the top three pathways, Myc targets, oxidative phosphorylation, and E2F targets were similar induced by aged human serum in vitro and in encapsulated myogenic progenitors in old mice. Counterintuitively, the category myogenesis was upregulated by aged serum in vitro after both four and ten days of exposure. Moreover, other gene set categories such as fatty acid metabolism that were strongly induced in both encapsulated cells in old mice and in aged freshly isolated myogenic progenitors, were not affected by aged serum in vitro.

After both four or ten days of exposure of hskMPs to aged human serum in vitro, we observed a robust response in downregulated gene sets (Figure 4—figure supplement 4b and Supplementary file 1m,n). However, in particular the top downregulated gene sets showed poor overlap with the profile obtained from encapsulated hskMPs or mskMPs in aged mice or in freshly isolated old myogenic progenitors. Moreover, myogenesis, which was consistently downregulated in both encapsulated cells and in freshly isolated in the aged condition, was not repressed by aged serum in vitro.

Importantly, in contrast to our observations using encapsulated cells, we observed that aged human serum in vitro did not increase the abundance of senescent β-galactosidase positive cells or affect the fusion index of the cells (Figure 4—figure supplement 3a-f and 5a-f). Overall, we demonstrate that the effects of systemic aging observed in encapsulated myogenic progenitors in old mice, can only partially be reproduced in vitro. These observations support the notion that certain systemic factors have a very short half-life or are not stable under cell culture conditions and can only be detected in the flux of the circulation.

To adapt the manuscript to these new broader conclusions, we decided to change its title to "In-Vivo Transcriptomic Profiling using Cell Encapsulation Identifies Effector Pathways of Systemic Aging".

b) Is the assay predictive: At the moment, the evidence that Myc and E2F are induced in aged myogenic cells is based on transcriptomic studies, and the conclusion that this aberrant signaling is driven by inflammatory processes is based on correlation to sera profiling. Further, validation of Myc / E2F as elements of an aging signature is based on comparison of comparing the transcriptomic profiling of myogenic progenitors culture in the capsules to freshly isolated muscle stem cells and brain tissue. Additional modes of validation are warranted to demonstrate the relevance of Myc / E2F in muscle cells and the predictive nature of the method.

The main goal of our paper is to provide a novel method for transcriptomic profiling of systemic aging and to provide an overview of the affected signaling pathways in myogenic cells. As outlined above, by performing an in vitro experiment using myogenic progenitors exposed to young and aged serum we validated Myc and E2F target genes as markers of systemic aging (Figure 4—figure supplement 4 and Supplementary file 1k-n). Importantly, in-depth functional validation of specific targets would take years of engineering, aging, and analysis of cell-type specific knockout mice. We hope the reviewer agrees that this goes beyond the scope of a single publication and that follow-up studies may be better suited for such a substantive undertaking. However, we'd like to highlight that our results fall well in line with the published literature. As mentioned in the discussion of our manuscript , it has been shown that mice haploinsufficient for Myc exhibit an increased lifespan and are resistant to osteoporosis, cardiac fibrosis and immunosenescence (Hofmann et al., Cell, 2015). Rapalogs, which inhibit the activity of mammalian target of rapamycin complex 1, increase life span and delay hallmarks of aging in many species. It has been shown that the rapalog RAD001, counter-regulates Myc in aged kidneys (Shavlakadze et al., J Gerontol A Biol Sci Med Sci, 2018). Moreover, E2F1, which interacts with the retinoblastoma tumor suppressor, has been shown to have a role in promoting cellular senescence (Dimri et al., Mol Cell Biol, 2000).

c) Is the assay optimized: The rationale for assay design criteria lack justification and therefore seem arbitrarily selected, which is unusual for a method paper. For example; (a) how was the analysis time-point selected to be Day 10?

As now explained in the Results section of our manuscript, differentiating hskMPs take about ten days until myonuclear acquisition is completed (Cheng et al., Am J Physiol Cell Physiol, 2014). In addition, based on the well-known timing of myogenic differentiation in vitro*,* we analyzed two time-points that cover early differentiation (4 days) and chronic adaptations in later stages (10 days) in the presence of young and aged serum (Figure 4—figure supplement 4 and Supplementary file 1 k-n). This in vitro experiment showed that after ten days a significantly higher number of pathways are affected than at the four day time point. Thus, in order to account for eventual differentiative effects of the systemic environment and to obtain a robust response in our profiling experiments, we decided to focus on the ten day time point for in vivo analysis of the capsules.

How does this choice relate to obtaining data that predicts aging factors that directly modulate muscle stem cells?;

To address this point, we compared the pathways affected in encapsulated myogenic cells in young and old mice to the aging profile of freshly isolated tissue resident muscle stem cells (Figure 4e). Pathways that are directly induced by the local aged niche microenvironment in freshly isolated MuSCs are those that do not overlap with the systemic aging profile obtained from encapsulated cells (i.e., bars that are not hatched in Figure 4e).

(b) was there a rationale in selecting a cell seeding density that is close to that used in skeletal muscle tissue engineering applications to create differentiated myotubes?;

The hollow fiber capsules we used for our experiments were optimized to be as small as possible allowing for adequate diffusion of long-range signaling factors, nutrients, gases, and ions and at the same time contain enough cells for downstream analysis. 10 µl containing 10k cells/µl of the cell-Matrigel mixture was the minimal concentration that allowed us to reproducibly obtain sufficient quality mRNA for bulk transcriptomics.

(c) with muscle stem cells as the desired comparator, were there desired criteria with regards to the differentiation status of the cells within the capsule?;

In fact, effects of the systemic environment on the balance of myogenic progenitor proliferation vs differentiation are not well studied and it was difficult for us to predict how the cells in the capsules would react. Primary myogenic progenitor culture is traditionally performed in rich artificial media containing only about 20% fetal bovine serum, while encapsulated cells in vivo are exposed to the constant flux of the systemic circulation. Therefore, to take possible pro-differentiative effects into consideration and let the encapsulated cells equilibrate, we decided to give them ten days before analysis, which, as mentioned above, is the time typically required to complete myonuclear acquisition in differentiating 2D culture (Cheng et al., Am J Physiol Cell Physiol, 2014).

(d) two reconstituted basement membrane (human and mouse) ECMs (~200 pascals) and TrueGel a non-adherent hydrogel (~10 kPa) were compared as scaffolds – what was the rationale for selecting these particular materials as it related to your experimental design?; (e) etc

Our goal was to identify a substrate that contains as little growth factors as possible allowing for an unbiased readout of systemic signals and at the same time keeping the cells viable. In addition, our selection of substrates was based on their commercial availability making it possible for researchers that want to implement our protocol to get easy access to the materials. Growth factor free Matrigel performed best with regard to these criteria (Figure 1—figure supplement 1b,c).

While the authors performed an elegant analysis to demonstrate that the myogenic progenitors remain viable and maintain stem cell attributes in capsules, the differentiation assay is based on fuzzy MyHC immunofluorescence and the stained cells do not show multinucleated phenotype as expected. Can the authors comment on the reason their cells did not fuse, or alternatively provide higher resolution images showing multinucleated fibers? This is of particular importance given that they refer to this assay as "terminal differentiation"- however do not provide sufficient evidence of multinucleated muscle formation.

As mentioned above, the effects of the systemic environment on the balance of myogenic progenitor proliferation vs differentiation are not well studied and it is possible that the cells in the capsules react differently from what we usually observe in 2D culture. In addition, it is important to keep in mind that the images we show are actually very thin cross sections of the cylindrical capsules. Thus, except myotubes that by chance differentiated exactly in the plane of the section, they will show up as oblong MyHC positive structures.

The authors need to validate using an additional assay aside from RNA-seq (i.e. RT-qPCR or immunofluorescence) the upregulation of select Myc or E2F targets in myogenic progenitors exposed to the aged mouse environment. On a similar note, they need to provide a full or at least partial list of these target genes.

To confirm our findings regarding Myc and E2F targets, we performed quantitative PCR using mRNA isolated from capsules containing hskMPs that were exposed to the systemic environment in young and aged mice for ten days. This experiment confirmed age-mediated induction of the Myc target small nuclear ribonucleoprotein polypeptide A' (Snrpa1) and the E2F target transferrin receptor (Tfrc) that were part of the respective gene sets upregulated in encapsulated cells in old mice (Figure 4—figure supplement 1c,d and Supplementary file 1d,e)

In addition, we performed an in vitro experiment using myogenic progenitors exposed to young and aged serum, which further validated Myc and E2F target genes as markers of systemic aging (Figure 4—figure supplement 4 and Supplementary file 1k-n).

We now also provide a list of Myc and E2F target genes in encapsulated cells after exposure to systemic aging in vivo (Supplementary file 1d,e).

Given the upregulation of targets of E2F it will be of interest to compare the expression level of canonical cell cycle regulators and senescent markers in myogenic progenitors exposed to old and young mouse environments. It is of further interest to assess if cells exposed to aged environment demonstrate less proliferation via EDU or Ki67 staining and upregulation of senescent markers.

To address this point we performed staining and quantification of the proliferation marker Ki-67 and the senescence marker β-galactosidase in encapsulated cells after in vivo exposure to a young and aged systemic environment (Figure 4—figure supplement 3a-d). While proliferation was not affected, the number of senescent cells was increased in encapsulated cells in the aged condition. Interestingly, cells exposed to young or aged serum in vitro did not show such changes (Figure 4—figure supplement 5a-d). Thus, exposure to an aged environment via in vivo encapsulation is sensitive to senescence, a well validated mechanism of geriatric conversion, while in vitro exposure to aged serum fails to recapitulate this important hallmark of aging.

The category "Myogenesis" is reported as one of the two primary downregulated clusters in myogenic cells exposed to an aged mouse environment. To corroborate this interesting observation, it will significantly strengthen the manuscript if the authors assess whether myogenic progenitors exposed to an old mouse environment exhibit reduced myogenic differentiation potential. This can be done via quantification of MyHC staining or analysis of myotube fusion index between cells exposed to the two conditions.

We performed a myosin heavy chain staining and quantified the fusion index of cells after encapsulation in young and aged mice (Figure 4—figure supplement 3e,f). This experiment confirmed a reduction of the myogenic capacity of the cells upon exposure to systemic aging. Once again, exposure to aged serum in vitro failed to recapitulate this phenotype (Figure 4—figure supplement 5e,f).

The RNA-Seq data analysis points to downregulation of inflammatory response in cells exposed to old vs. young systemic environment, however the authors also concomitantly report on elevated levels of systemic inflammation markers detected in blood of aged mice. Can the authors comment or discuss the discrepancy between these two opposing observations?

In the discussion of our paper, we elaborate regarding a role of Myc signaling in mediating inflammatory responses and cite Liu et al., (please see below for the full reference). Nevertheless, we agree that it is intriguing that in spite of higher levels of proinflammatory systemic cytokines, certain inflammatory processes appear to be downregulated as a consequence of exposure to systemic aging. It is possible that, compared to immune cells whose response to pro-inflammatory molecules is well characterized, stem cells react non-canonically.

Liu, T., Zhou, Y., Ko, K.S., and Yang, H. (2015). Interactions between Myc and Mediators of Inflammation in Chronic Liver Diseases. Mediators Inflamm 2015, 276850

The comparison to a previously published dataset from a parabiosis trial is an underdeveloped off-shoot and the message the authors wish to convey is supported by thin data analysis. The authors need to expand their analysis or discussion on the significance of this observation or alternatively consider removing it from the manuscript.

We removed the dataset from the paper.